# On Overcoming Miscalibrated Conversational Priors in LLM-based Chatbots

**Christine Herlihy**[1]      **Jennifer Neville**[2]      **Tobias Schnabel**[2]      **Adith Swaminathan**[2]

[1]Department of Computer Science, University of Maryland, College Park, MD USA
[2]Microsoft Research, Redmond, WA USA

## Abstract

We explore the use of Large Language Model (LLM-based) chatbots to power recommender systems. We observe that the chatbots respond poorly when they encounter under-specified requests (e.g., they make incorrect assumptions, hedge with a long response, or refuse to answer). We conjecture that such miscalibrated response tendencies (i.e., conversational priors) can be attributed to LLM fine-tuning using annotators — single-turn annotations may not capture multi-turn conversation utility, and the annotators' preferences may not even be representative of users interacting with a recommender system. We first analyze public LLM chat logs to conclude that query under-specification is common. Next, we study synthetic recommendation problems with configurable latent item utilities, and frame them as Partially Observed Decision Processes (PODP). We find that pre-trained LLMs can be sub-optimal for PODPs and derive better policies that clarify under-specified queries when appropriate. Then, we re-calibrate LLMs by prompting them with learned control messages to approximate the improved policy. Finally, we show empirically that our lightweight learning approach effectively uses logged conversation data to re-calibrate the response strategies of LLM-based chatbots for recommendation tasks.

## 1 INTRODUCTION

In contrast to their task- or domain-specific predecessors, modern conversational agents have employed large language models (LLMs) to achieve high proficiency levels (i.e., at or exceeding that of humans) in challenging, open-domain settings [Achiam et al., 2023]. The implicit objective for the agent in such settings is to respond to a user in a way that

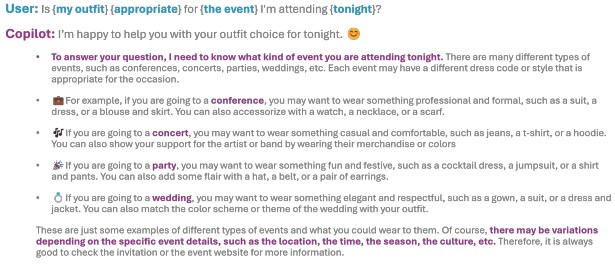

Figure 1: An example failure where a user's query is under-specified (blue text). Current LLM-based chatbots produce long responses in order to hedge against uncertainty (purple text). Clarifying the user's context can avert this failure.

maximizes the user's utility given their conversation goal(s).

However, humans are often unable or rationally unwilling to fully verbalize (i.e., explicitly state) their goals and preferences for various reasons (e.g., efficiency) and may instead rely on their conversational partner(s) to fill in the gaps [Piantadosi et al., 2012]. This leads users to issue *under-specified* queries in which the LLM-based chatbot observes only a subset of the preferences and constraints required to provide a high-quality answer – see Figure 1 for an example. Empirically, we observe that under-specification is common: we classified a random sub-sample of the queries in the OpenAssistant dataset [Köpf et al., 2023] and found that more than 23% of queries posed to LLM-based chatbots today are severely under-specified (see Figure 2 and Section 3.1 for details).

In this paper, we explore the relationship between query under-specification, LLM response behavior, and user satisfaction. We begin by proposing a taxonomy of LLM response strategy types (see Table 2) to characterize the behavior of SoTA models in the face of query under-specification— i.e., their "conversational priors"— with respect to utility and cognitive cost [Tankelevitch et al., 2023]. Figure 3 provides a demonstrative example. Note that each response

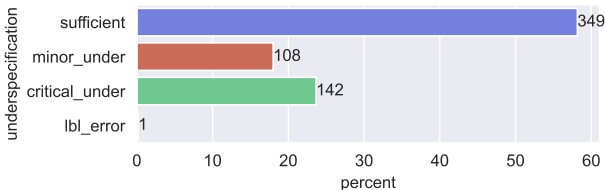

Figure 2: Real-world users asked severely under-specified queries more than 23% of the time in the OpenAssistant dataset ($n = 600$).

**Q: "Is my outfit appropriate for the event?"**

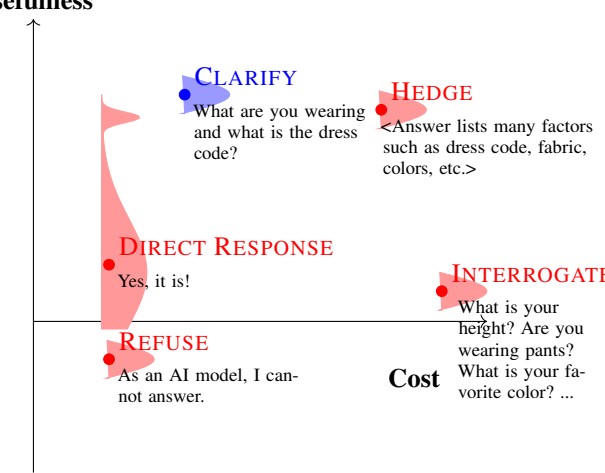

Figure 3: For a user query such as $q$: "Is my outfit appropriate for the event I'm attending tonight?" an LLM-based chatbot can choose different response strategies. These strategies produce responses that differ in their cognitive costs (x-axis) while providing final answers with different, user-specific levels of usefulness (y-axis). A good chatbot should respond so as to maximize overall utility—i.e., by providing useful and low-cost answers for the user.

strategy (a) can be characterized by syntactic and semantic features (i.e., length, presence or absence of conditional statements/questions, etc.) and (b) will give rise to a joint distribution over cost and utility that impose different trade-offs depending on the user's true but latent preferences.

We use this taxonomy and a combination of synthetic and real-world queries to empirically demonstrate that: (a) SoTA LLMs are predisposed to respond directly or hedge in lieu of asking a small number of clarifying questions when queries are under-specified; and (b) such miscalibration can lead to unsatisfactory and/or sub-optimal performance on downstream tasks (as illustrated in Figure 1 and Section 3.2).

To address the miscalibration of LLMs outlined above, we formalize user-chatbot interactions as a partially observable decision process (PODP), where a user with a partially ob-

servable goal engages in a turn-by-turn conversation with a chatbot. In this PODP, the chatbot's policy $\pi$ is a fixed mapping from conversation prefixes (which can span multiple turns) to natural language responses. Then, for any given conversation and user goal, the chatbot seeks to provide a natural language response that maximizes utility according to a fixed but unknown user utility function. Note that utility is computed with respect to the user's *latent* goal, which may be *fully* or *partially* observable via their query.

Intuitively, when the goal is partially observable and the user is amenable to answering a small number of clarifying questions, a policy that produces a natural language response containing questions at timestep $t_0$ and incorporates the information gained to produce higher-quality responses at future timestep(s) will yield higher expected *cumulative* utility, relative to a myopic policy that tends to respond directly or hedge at $t_0$. We build upon this insight to propose two interventions (Sections 4.1 and 4.2) to make LLM-based chatbots produce better-calibrated responses in the face of query under-specification. Both of the interventions require only API access to frozen, black-box LLMs.

Our first intervention (Section 4.1) is inspired by prior research on the generation of clarification questions Rao and Daumé III [2018], Majumder et al. [2021], and uses a static, "clarification-aware" prompt to nudge LLMs to clarify when appropriate rather than reverting to default response behavior. Our second intervention (Section 4.2) leverages historical conversation logs to learn a meta-policy—i.e., a mapping from conversation prefixes to a finite set of prompts. Then during a PODP episode, the chatbot first invokes this meta-policy, and then calls the LLM with the resulting prompt to produce a contextually appropriate PODP action. We expect the two proposed interventions to be effective in different data regimes — if high-quality logged data is readily available, the approach in Section 4.2 is a practical alternative to resource-intensive approaches such as fine-tuning LLMs on the collected data. Conversely, if we do not have access to sufficient high-quality data, we may prefer the data-agnostic approach of Section 4.1.

In Section 6, we highlight that our proposed interventions can be further improved—for instance, reasoning about "good" clarification questions to ask (currently left up to the LLM) and the propensity of users to answer with relevant information. Empirically, we evaluate both interventions on recommendation tasks featuring a synthetic user model. We find that each intervention achieves higher expected utility relative to baseline when queries are under-specified, and converges to baseline as query specification increases.

## 2  PROBLEM FORMULATION

In the PODP setting that we consider, let $\theta \in \Theta$ represent a user's latent, conversation-level *goal*. Each PODP

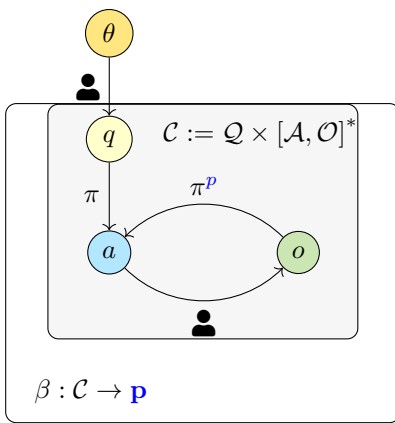

Figure 4: PODP plate diagram illustrating user-chatbot interactions, prompt-induced policies ($\pi^p$), and the meta-policy mapping from conversations to prompts ($\beta$).

*episode*—i.e., user-chatbot conversation—begins with the user expressing their goal, $\theta$, in a potentially lossy manner via a natural language query, $q \in \mathcal{Q}$. Per Definition 2.1, we consider a query $q$ to be *under-specified* if there is an information gap between the user's goal and stated query:

**Definition 2.1** (Under-specification)**.** Query under-specification is the partial observability of user's goal given a query, i.e., $\Pr(\Theta \mid \mathcal{Q})$ is unknown and not deterministic.

Table 1 lists some examples of under-specified queries in the OpenAssistant dataset (see Section 3.1 for details).

Once initiated, a conversation dialogue is assumed to proceed iteratively until terminated by the user (Figure 4). In this context, the chatbot's natural language responses constitute the *action space* of the PODP and are denoted by $a \in \mathcal{A}$, while the user's follow-up utterances constitute *observations* denoted by $o \in \mathcal{O}$. We denote the multi-turn, variable-length *conversation history* between the user and chatbot by $C \coloneqq q \times [a, o]^*$. We use $\mathcal{C} \coloneqq \mathcal{Q} \times [\mathcal{A}, \mathcal{O}]^*$ to refer to the space of conversation histories. Then, for any chat conversation $C$ with user goal $\theta$, the task of the chatbot system is to produce actions with maximum utility according to a fixed but unknown *user utility function*, $\mathcal{U} : \Theta \times \mathcal{C} \mapsto \mathbb{R}$. Although the reward function of the PODP, $\mathcal{U}$, is unknown we can observe samples from it. For example, many LLM-based chatbots allow users to rate their conversations; these ratings can be directly interpreted as $\mathcal{U}(\Theta, C)$. Recent work [Lin et al., 2024] infers $\mathcal{U}$ across a user population using a small sample of rated conversations. In general $\mathcal{U}$ can rely on a mix of implicit factors, such as response length, and explicit factors, such as thumbs up/down or user ratings of paired responses. Moreover, in Figure 3 we saw that the cognitive cost imposed on the user can be another component influencing $\mathcal{U}$; in our experiments, we use response length—$\texttt{len}(a)$—as a simple proxy for a user's cognitive cost from action $a$.

We define the *policy* $\pi$ of a chatbot interacting with a user as a stationary (but not necessarily Markovian) mapping from conversation histories to natural language responses $\pi : \mathcal{C} \to \mathcal{A}$ (Figure 4). An optimal chatbot policy is one that maximizes expected utility:

$$\pi^* \approx \arg\max_{\pi} \mathbb{E}_{\{\theta,q\}} \mathbb{E}_{a\sim\pi}[\mathcal{U}(\theta, q, [a, o]^*, a)]. \quad (1)$$

In Equation 1, note that the policy influences the responses $a$ in all turns of the conversation, and that $\Pr(\Theta, \mathcal{Q})$ is sampled from the user population.

## 2.1 POLICIES INDUCED BY PROMPTING LLMS

System messages (also known as *prompts*) are often used to "steer" an LLM and induce specific behaviors (e.g., $\mathbf{p}$ ="Behave as a helpful assistant"). For LLMs that do not support a separate system message $\mathbf{p}$, the prompt and conversation transcript can be concatenated together into the LLM's input context $\coloneqq \mathbf{p} \circ C$. Otherwise, PODP policies can be induced by using a prompt $\mathbf{p}$ and LLM input context $\coloneqq C$. Such PODP policies are denoted as $\pi^{\mathbf{P}}$. If we restrict our attention to the chatbot policies we can access via prompting, we can rephrase the *policy* optimization objective (i.e., Equation 1) in terms of *prompt* optimization:

$$\mathbf{p}^* \approx \arg\max_{\mathbf{p}} \mathbb{E}_{\{\theta,q\}} \mathbb{E}_{a\sim\pi^{\mathbf{P}}}[\mathcal{U}(\theta, q, [a, o]^*, a)]. \quad (2)$$

When we implement a policy by querying a blackbox LLM API with context $\coloneqq C$ (i.e. $\mathbf{p}$ is empty), we refer to the induced PODP policy as the RLHF policy $\pi^{\text{RLHF}}$. We can expect good PODP performance out-of-the-box from an LLM only if its RLHF-finetuning guarantees that $\pi^{\text{RLHF}} \approx \pi^*$ (which is unverifiable).

## 2.2 QUERY UNDER-SPECIFICATION CAUSES SUB-OPTIMAL INTERACTIONS

Modern LLMs are typically fine-tuned via RLHF, where the training objective [Ouyang et al., 2022] corresponds to:

$$\pi^{\text{RLHF}} \approx \arg\max_{\pi} \mathbb{E}_{\{\theta,q\}\sim\text{lab}} \mathbb{E}_{a\sim\pi}[\mathcal{U}(\theta, q, a)]. \quad (3)$$

The combination of query under-specification and RLHF fine-tuning impacts policy learning (i.e., via Equation 3) in two ways: (1) distribution shifts between the preferences of annotators and those of end-users may skew the learned policy; and (2) RLHF's emphasis on annotation of, and optimization over, *single-turn* interactions produces myopic policies that greedily maximize single-turn utility.

With respect to (1), annotators may not be able to reliably infer users' *true* preferences (i.e., $\theta$) when evaluating possible responses to user queries—i.e., $\Pr_{\text{lab}}(\Theta \mid \mathcal{Q}) \neq \Pr(\Theta \mid \mathcal{Q})$.

Additionally, the utility function may also shift. For example, Singhal et al. [2023] observe that RLHF annotators may prefer longer, more detailed responses relative to end-users.

With respect to (2), the focus on single-turn interactions means annotators are less likely to be exposed to conversations where a chatbot asks the user clarification questions to better understand and respond to the user's query, because such conversations will, by definition, require multiple turns. In the single-turn setting, annotators may also perceive responses that attempt to answer users' queries (albeit incorrectly or verbosely) as more *helpful* than responses containing clarification questions. Policy learning with such preferences may thus underestimate the value of uncertainty-reducing behaviors such as clarification, and the resulting policy may be sub-optimal for *multi-turn* conversational outcomes in PODPs. We empirically show that these challenges render $\pi^{\text{RLHF}}$ sub-optimal compared to $\pi^*$.

## 2.3 META-POLICIES

When prompting LLMs to produce chatbot responses, we are not limited to using a fixed prompt for all conversation turns. Instead, we can define a meta-policy, $\beta : \mathcal{C} \mapsto \mathbf{p}$ as a mapping from conversation prefixes to prompts. A PODP agent acting during an episode can first invoke the meta-policy $\beta$, and then query the LLM with prompt $\mathbf{p} := \beta(C)$ to produce its action. For PODP policies implemented through a composition of a meta-policy with an LLM, the original problem of finding a good $\pi^*$ is replaced with finding a good *meta-policy* $\beta^*$:

$$\beta^* \approx \arg\max_{\beta} \mathbb{E}_{\{\theta,q\}} \mathbb{E}_{a \sim \pi^{\mathbf{P}}} [\mathcal{U}(\theta, q, [a, o]^*, a) \mid$$

$$\mathbf{p} = \beta(q, [a, o]^*)].$$

Note that learning a meta-policy $\beta$ is a *different* decision-making problem than the PODP decision-making problem (i.e., action space of prompts instead of chatbot responses).

## 2.4 CHARACTERIZING AND INDUCING CHATBOT RESPONSE BEHAVIORS

To empirically evaluate $\pi^{\text{RLHF}}$ and to design prompt-based interventions, we introduce a taxonomy (detailed in Section 3.2) that can be used to (1) characterize LLM response behavior; and (2) constrict the meta-policy's action space. Regarding (1), we refer to the distribution of response strategies of $\pi^{\text{RLHF}}$ as the LLM's "conversational prior" (e.g., see Figure 6 for GPT-4's conversational prior).

To build intuition for how this taxonomy may serve both purposes, note that $\pi^{\text{RLHF}}$ can be viewed as a hierarchical probabilistic process in which the chatbot first samples a latent response strategy, $\tau \sim \mathcal{T}$, and then generates a natural language response conditioned on the response strategy,

Figure 5: Spectrum characterizing the response strategies that a LLM-based chatbot can take. RLHF fine-tuning encourages RESPOND and HEDGE, whereas CLARIFY may be more appropriate when user queries are under-specified.

$a \mid \tau$. Then, if $\pi^{\text{RLHF}}$ is found to be miscalibrated in its distribution over $\mathcal{T}$, we can *intervene* via prompts to promote desired response behavior(s).

We specifically consider a set of response strategies, $\mathcal{T} = \{\text{REFUSE, RESPOND, HEDGE, CLARIFY, INTERROGATE}\}$. To motivate this choice, recall that in the PODP, the chatbot cannot observe the user's intent, $\theta$, and must instead act based on the *belief state*—i.e., $\Pr(\theta \mid q, [a, o]^*)$. In this context, possible response strategies lie along a spectrum characterized by the relative *absence* or *presence* of (belief)-uncertainty-reducing behavior(s) (Figure 5).

On the *uncertainty-agnostic* end of this spectrum, the chatbot may rely on its inductive prior to *respond directly*—i.e., despite uncertainty about the user's preferences. Responding directly relies on assumptions and/or potentially spurious semantic correlations between the preferences the user *does* express and those that the LLM-based chatbot must infer. On the *uncertainty-reducing* end, a chatbot may ask an unbounded number of questions before responding (*Interrogate*). This can allow the system to best approximate a user's fully specified intent but is completely irrational for the user to engage with. As Figure 3 shows, any deviations from the *Respond* response strategy must be done in a thoughtful manner, lest the user have a worse cost-utility benefit even as the system reduces uncertainty in its beliefs.

In a PODP, it is critical to balance information-seeking (exploration) against utility maximization (exploitation). In Section 3, we demonstrate that $\pi^{\text{RLHF}}$ places too much weight on response strategies that myopically maximize one-step utility (i.e., RESPOND and HEDGE). In Section 4.1, we demonstrate that a simple prompt is able to shift the distribution over response strategies toward CLARIFY when queries are under-specified, and thereby improve the PODP policy.

## 3 MOTIVATING EXPERIMENTS

Here, we establish that: (1) query under-specification is common in real-world human-chatbot conversations; and (2) $\pi^{\text{RLHF}}$ can be sub-optimal when queries are under-specified.

## 3.1 QUERY UNDERSPECIFICATION IS COMMON

We annotated the OpenAssistant dataset [Köpf et al., 2023] to explore how often users issue under-specified queries to open-domain LLM-based chatbots. We restrict our study to queries in English with at least 3 words ($\approx 40\%$ of over

10, 000 conversations) and subsample 600 queries uniformly at random. We created an LLM-based classifier to map each query to a predicted under-specification label, whose accuracy we also validate on a synthetic corpus (see Appendix A.1). Class labels include:

- CRITICAL UNDER: One or more important factors upon which an answer to this query might depend are not specified or are unknown; it is difficult to provide a high-quality response without knowing these factors.

- MINOR UNDER: Less important factors that the query might depend on are not specified or are unknown; however, it is possible to provide a high-quality response even without knowing these factors.

- SUFFICIENT: All important factors upon which an answer to this query might depend are sufficiently specified.

Figure 2 summarizes the results of this experiment, which shows that query under-specification is prevalent. A few examples of critically under-specified queries are listed in Table 1. Note also that many OpenAssistant users have experience with prompting, and we conjecture a higher prevalence of under-specified queries from novice user populations.

| Critically Under-specified Queries (Abridged) |
|---|
| Suggest me places near 72nd St where I can park my car. |
| What are some up and coming and high quality youtube channels in science and technology that I have probably not heard of? |
| A friend of mine barely talks to me anymore and I don't know why. |

Table 1: Examples from the OpenAssistant dataset tagged by our classifier (details in Appendix A.1).

## 3.2 LLM POLICIES CAN BE SUB-OPTIMAL WHEN QUERIES ARE UNDER-SPECIFIED

When queries are under-specified, $\pi^{\text{RLHF}}$ has difficulties optimally trading-off information seeking with greedy, utility-maximizing response tendencies. To study this, we define seven broad categories for query responses in Table 2. We use these definitions with an LLM-based classifier, which we validate in Appendix A.2. Let $\tau$ be the predicted response type of response $a$. Our experiments show that both for real-world and synthetic queries, the current SoTA LLM, GPT-4, prefers to either directly respond or hedge, instead of clarifying via a short question.

### 3.2.1 Synthetic query corpus

The goal for the synthetic corpus is to have a full-information setting where we can explicitly control the degree of under-specification and measure the utility of any given response. We generate queries for three different recommendation domains (movies, gifts, plants) that each have four constraint dimensions $\theta_i$ that can be active

| Response type $\tau$ | Response characteristics |
|---|---|
| REFUSE | Contains an explicit or implicit refusal to answer. |
| DIRECT RESPONSE | No questions or hedging; addresses query. |
| HEDGE | Many answers, conditioned on uncertain factors. |
| CLARIFY | Limited/prioritized set of questions (i.e., $\leq 3$). |
| INTERROGATE | Large/exhaustive number of questions (i.e., $> 3$). |
| MISSING | The response is empty/blank. |
| MISCELLANEOUS | Describes or follows query instructions. |

Table 2: For the motivating experiments in Section 3, we categorize LLM responses into seven response types.

(set to a specific value, e.g., $\theta_{age}$ = "25-35 years"), or inactive, (e.g., $\theta_{age} = \emptyset$). We base this setup on Radlinski et al. [2019], who studied users' preferences for movies expressed in a conversational recommendation setting. The user goal is then to get a recommendation that satisfies *all* of these constraints. Constraint values and the number of active dimensions are sampled via uniform sampling. After determining the ground truth user goal $\theta$, we generate a potentially under-specified user query by sampling a subset of active constraint dimensions to reveal. With a slight abuse of notation, let $q$ be the vector of revealed active constraints. We categorize the resulting queries as:

$$q \mapsto \begin{cases} \text{CRITICAL UNDER} & |q| \leq 1, \\ \text{SUFFICIENT} & |q| = |\theta|, \\ \text{MINOR UNDER} & \text{otherwise.} \end{cases} \quad (4)$$

Details can be found in Appendix C.1.

### 3.2.2 Sub-optimality of LLM in single-step interaction

For each query, we use GPT-4 with the default system message to generate a natural language response $a \sim \pi^{\text{RLHF}}$ and assign it a response type label $\tau$ from Table 2 using our LLM-based classifier.

Figure 6 shows the distribution over response strategies (by corpus and under-specification severity) for $\pi^{\text{RLHF}}$. We observe that for both synthetic and real-world queries, using the uncertainty-agnostic DIRECT RESPONSE strategy is preferred by a large margin across *all* under specification buckets. While there is evidence that uncertainty-aware response strategies (i.e., HEDGE, CLARIFY, and INTERROGATE) are increasingly used when under-specification rises, the sheer magnitudes still express a clear bias for $\pi^{\text{RLHF}}$ to respond or hedge—rather than clarify—in the face of under-specification. This indicates that there is headroom to improve utility even over SoTA LLMs.

### 3.2.3 Sub-optimality of LLM in multi-step interactions

Intuitively, a policy asking a few relevant questions in the beginning should be able to outperform $\pi^{\text{RLHF}}$ in many cases since $\pi^{\text{RLHF}}$ often defaults to DIRECT RESPONSE. The following two-step recommendation task shows this.

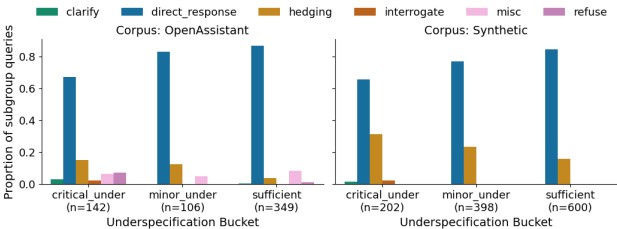

Figure 6: Even under severe levels of under-specification, GPT-4 prefers to directly answer a user query.

We compare $\pi^{\text{RLHF}}$ with two simple static policies described in Table 3. We use modified system messages to encourage different behavior for the first response, and follow $\pi^{\text{RLHF}}$ as the default policy after (for full prompts, see Appendix B).

| Policy $\pi^{\text{P}}$ | System prompt $\mathbf{p}_0$ |
|---|---|
| $\pi^{\text{RLHF}}$ | Default LLM system message (unmodified). |
| $\pi^{\text{Clarify}}$ | Ask about $\leq 3$ of *most relevant* factors. |
| $\pi^{\text{Hedge}}$ | Condition on option(s) for each uncertain factor. |

Table 3: We evaluated three different policies that encourage different initial response strategies to show the possible room for improvement in multi-step interactions.

We use the queries and ground truth user goals from the synthetic query corpus outlined in Section 3.2.1, but focus on the movie domain only, following Cheng et al. [2023]. Each episode begins at $t = 0$ (denoted $t_0$) with the user issuing query $q$ to ask for movie recommendations that satisfy their true preferences $\theta$. When the LLM-based chatbot provides recommendations (i.e., chooses action types DIRECT RESPONSE or HEDGE), we terminate the episode and compute the utility of the recommendation. If the chatbot asks questions, we use another LLM as a user simulator, requiring the latter to divulge information in a templatized format about constraints $\theta_i$ *only* if explicitly asked (see Appendix C.4).

**Item Utilities.** We begin by measuring the utility of items recommend by each $\pi$, operationalized as the fraction of constraints (out of $4$) that an item satisfies, averaged across all items recommended to the user. Instead of comparing individual policies, we compare response types $\tau$ to eliminate cases where setting the system message $\mathbf{p}$ did not induce the desired response type. Figure 7 shows how multi-step episode utilities develop when we group by the type of the first system response, $\tau_0$. CLARIFY does not generate any utility at time $t_0$, since no recommendations have been made, but does much better in the second time step $t = 1$, especially for critically under-specified queries. When we HEDGE in the beginning, we do get utility at $t_0$, but generate less than when we directly reply, since utility is averaged over *all* (possibly irrelevant) recommendations.

These findings suggest that there is headroom for improvement over $\pi^{\text{RLHF}}$ in multi-step interactions.

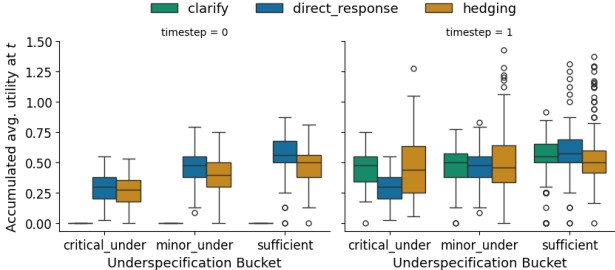

Figure 7: Distribution of accumulated item utilities $\mathcal{U}$ at timesteps $t = 0, 1$; grouped by under-specification levels.

**Costs.** We now consider how the *cost* of capturing this headroom—i.e., moving from an under-specified query to a more fully specified version—varies over the uncertainty-aware strategies that we consider—i.e., HEDGE and CLARIFY. To proxy for the cognitive burden associated with reading and answering clarification questions or parsing the many cases or conditions mentioned in hedging responses, we define a cost function, $c : \mathcal{A} \to \mathbb{R}_{\geq 0} := \text{len}(a)$ (measured by counting all unigrams in $a$).

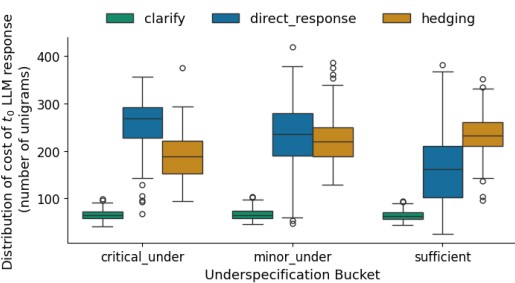

Figure 8: Distribution of response cost at $t = 0$ for each response strategy $\tau_0$; grouped by under-specification levels.

Figure 8 illustrates the benefit of CLARIFY – it carries a relatively low cost in terms of output length. Interestingly, the DIRECT RESPONSE action produces the longest answer among all other response types when queries are critically under-specified. Inspecting the produced responses, we see that DIRECT RESPONSE produces long answers by adding explanations or extended lists of recommendations. When queries are sufficiently specified, HEDGE leads to the highest cost answers, as it still enumerates over many answer options. Overall, we see that CLARIFY obtains the lowest average cost across *all* under-specification buckets, suggesting that a policy could achieve higher utility with lower costs by considering the CLARIFY action more often.

## 4 ALGORITHMIC APPROACH

In this section, we outline two algorithmic interventions to improve upon $\pi^{\text{RLHF}}$ in PODPs. The first intervention uses a fixed prompt to an LLM-based policy $\pi^{\text{P}}$ that nudges the

LLM to prefer cost-aware uncertainty-reducing response strategies like clarifications when appropriate. We saw in Section 3 that this *data-agnostic* approach can be substantially better than $\pi^{\mathrm{RLHF}}$ when queries are under-specified and users patiently respond to all clarifications. However, real-world users may have varying propensities to engage with clarifying questions. So, we devise a second intervention in Section 4.2 that uses historical conversational logs to fit an appropriate *meta-policy* $\beta$ that can be more optimal for the PODP.

## 4.1 DATA-AGNOSTIC INTERVENTIONS

We saw in Section 3 that LLM-based chatbots have sufficient capabilities at *detecting* under-specified queries (Section 3.1) and generating CLARIFY responses if prompted explicitly (Section 3.2.3). However, they do *not* appear to sufficiently condition on their latent under-specification judgments when generating responses in the absence of intervention (i.e., when relying on the baseline system message in $\pi^{\mathrm{RLHF}}$). Thus, we consider two approaches that explicitly emphasize the possibility of under-specification and the benefits of clarification when appropriate and allow graceful recovery of default system behavior when warranted—e.g., when queries are well-specified.

**Approach 1: Chain of Thought (CoT).** We evaluate a chain-of-thought [Wei et al., 2022] intervention in the form of a modified system message that encourages the LLM-based chatbot to "ask yourself whether you have sufficient information to provide a good answer, and then respond accordingly" when responding to queries (see Appendix B).

**Approach 2: Clarify When Appropriate (Clarify-Flex).** We also evaluate a more flexible, context-aware relaxation of the "always clarify" system message that we experimented with in Section 3.2.3. This modified system message instructs the LLM-based chatbot to ask clarifying questions about important factors only *if* they have not been specified, and to respond directly otherwise (see Appendix B).

**Key Findings and Limitations.** In order to compare our data-agnostic interventions to $\pi^{\mathrm{RLHF}}$, we conduct a slightly modified version of the two-step recommendation experiment presented in Section 3.2. Here, we consider $\mathbf{p}_0$ values $\in \{\textsc{Baseline}, \textsc{CoT}, \textsc{ClarifyFlex}\}$, and sequential combinations $\in \{(\mathbf{p}_0, \mathbf{p}_1) \mid \mathbf{p}_1 = \mathbf{p}_0 \vee \mathbf{p}_1 = \textsc{Baseline}\}$.

We begin by using our LLM-based $\tau$-classifier to map each intervention to a distribution over response strategies, so as to assess the extent to which highlighting uncertainty and encouraging contextual awareness at response generation time induces changes in response behavior relative to baseline. As Figure 9 illustrates, while the CoT intervention behaves quite similarly to the BASELINE, CLARIFYFLEX meaningfully diverges, favoring *interrogation* when queries

are critically under-specified, then shifting toward *clarify*, and finally toward *direct response* (i.e., converging with BASELINE) as the degree of specification increases.

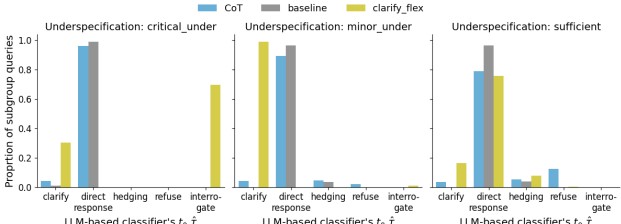

Figure 9: Distribution of the response strategies $\hat{\tau}_0$ induced by the three prompts $= \{\textsc{Baseline}, \textsc{CoT}, \textsc{ClarifyFlex}\}$; grouped by under-specification levels.

Next, we examine the distribution over the average utility of recommended items for each sequential combination of $(\mathbf{p}_0, \mathbf{p}_1)$. As Figure 10 illustrates, (CLARIFYFLEX, BASELINE) is the best-performing sequential combination when queries are critically under-specified, with relative advantage diminishing as specification increases. When queries are sufficiently specified, (CLARIFYFLEX, BASELINE) and (CoT, BASELINE) obtain slightly higher median $\bar{\mathcal{U}}$ than (BASELINE, BASELINE), but we generally see convergence due to the fact that both baseline and interventions tend toward direct response in this setting.

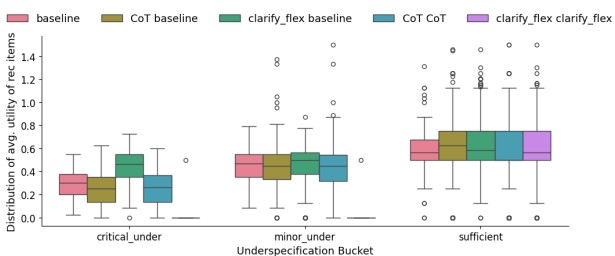

Figure 10: Distribution of $\bar{\mathcal{U}}$ for each $(\mathbf{p}_0, \mathbf{p}_1)$ sequence; grouped by under-specification levels.

From this analysis, we conclude that among the data-agnostic interventions we consider, CLARIFYFLEX is best able to improve upon the baseline $\pi^{\mathrm{RLHF}}$ when queries are critically under-specified, while maintaining the flexibility to converge to *direct response* as specification increases. In summary, through Figures 9, 10, we see that in a synthetic user model (that provides templatized answers to clarification questions), it is possible to improve upon the performance of the baseline LLM—i.e., CLARIFYFLEX performs better than $\pi^{\mathrm{RLHF}}$ when evaluated in the PODP.

## 4.2 DATA-BASED INTERVENTION

Here, we introduce an intervention that leverages collected conversation logs to learn *when* and *how* to improve upon

$\pi^{\text{RLHF}}$—i.e., by redistributing probability mass away from uncertainty-agnostic *direct response* and cost-agnostic *hedging* toward cost- and context-aware response strategies such as *clarify* when appropriate—in a way that is more tunable and adaptive to different user populations than the data-agnostic interventions we consider in Section 4.1.

We begin by considering meta-policies $\beta$ as described in Section 2.3. Remember that learning a mapping $\beta : \mathcal{C} \mapsto \mathbf{p}$ is a *different* decision-making problem than the original PODP policy. As described in Section 2.4, we will use the taxonomy we developed in Table 2 to reduce the action space of the meta-policies. Given a $\mathcal{T}$ with corresponding prompts $\mathbf{p}_\tau : \tau \in \mathcal{T}$, we consider the restricted set of meta-policies $\beta : \mathcal{C} \mapsto \mathcal{T}$. A PODP agent using $\beta$ will, at each timestep, first calculate $\hat{\tau} = \beta(\mathcal{C})$, look up the corresponding prompt $\mathbf{p}_{\hat{\tau}}$ and finally query the LLM with $(\mathbf{p}_{\hat{\tau}}, \mathcal{C})$ to produce an action in the PODP.

Conceptually, if we had the ability to simulate the PODP environment, then we could learn a meta-policy $\beta$ through online Reinforcement Learning (RL): i.e., sample prompts at each turn in the conversation from the current $\beta$, observe the resulting conversation-level outcomes, and update the parameters of $\beta$ using e.g., PPO. However, we typically cannot simulate user-chatbot conversations with high fidelity, and running online RL with users directly can be very sample inefficient and result in a poor user experience.

Instead, we use an offline approach inspired by Asymmetric Imitation Learning [Pinto et al., 2018]. We assume access to a dataset $\mathcal{D}$ containing logs of user-chatbot dialogues along with conversation-level utility ratings, $\mathcal{D} = \{(C_1, U_1) \ldots (C_n, U_n)\}$. Such a dataset can be collected, for example, from an already deployed chatbot. Notice that the data contains signals about the true $\theta_i$ (i.e. $U_i := \mathcal{U}(\theta_i, C_i)$) beyond what can be inferred from $C_i$, but the learner $\beta$ does not have access to $\theta_i$. Hence, imitating optimal actions in $\mathcal{D}$ reduces to asymmetric imitation learning.

We use the $\tau$-classifier developed in Section 3.2 to annotate all of the chatbot responses in $\mathcal{D}$ with their response strategy $\hat{\tau}$. Consequently, we can estimate a Q-value function $Q(C, \hat{\tau})$ on the annotated data as:

$$\hat{Q} = \arg \min_Q \sum_{i \in \mathcal{D}} \sum_{a_j \in C_i} (Q(C_i[: a_j], \hat{\tau}(a_j)) - U_i)^2,$$

where $C[: a]$ denotes the conversation prefix upto the chatbot response indicated by $a$. The Q-value function $\hat{Q}(C, \tau)$ estimates the eventual utility the learner will receive if we take action $\tau$ upon observing conversation $C$ and then follow the baseline system (i.e., $\pi^{\text{RLHF}}$) at all future timesteps.

When new conversations arrive, we evaluate the predicted $Q$ values for each $\tau \in \mathcal{T}$ and choose the argmax:

$$\beta(C) = \arg \max_{\tau \in \mathcal{T}} \hat{Q}(C, \tau). \tag{5}$$

We empirically evaluate this Q-value estimation approach in the synthetic recommendation experiment. We operationalize reward as the average utility (i.e., alignment between an item's features and the user's true preferences) over the set of recommended items. In the synthetic setup, we can generate responses (and eventual conversation rewards) for all possible $\tau \in \mathcal{T}$ for each query seen in the dataset $\mathcal{D}$. So we compute $Q^*$ for all queries seen in $\mathcal{D}$. However we need to estimate $Q$ for new queries as they arrive so as to implement Equation 5.

We construct a regressor to estimate $Q^*$ as follows: we use a pre-trained SentenceTransformer model [HuggingFace, 2024] to encode a stratified sample of our synthetic corpus (we stratify by the degree of under-specification so that the resulting distribution over labels mimics the OpenAssistant results we report in Figure 2).

Then, for new conversation histories, e.g. $q$, we encode it using the same embedding model and retrieve its $k$-nearest neighbors, with $k = 5$. We then retrieve each neighbor's $Q^*$ and corresponding $\tau$. We can then predict the Q-value of each candidate $\tau$ as the average of the $Q^*(\tau)$ values contributed by neighbors. This is akin to an asymmetric imitation learning baseline [Sinclair et al., 2023]. We greedily choose the argmax $\tau$ at $t_0$, simulate user answers to LLM responses containing questions as in Section 3.2.3, follow $\pi^{\text{RLHF}}$ at $t_1$, and report the resulting episode-level rewards (i.e., average utility over items in the rec set). We present empirical results for this approach in Figure 11, and observe that our learned meta-policy achieves higher reward relative to baseline. The empirical results demonstrate that both strategies we evaluate—i.e., *designing good prompts* (Section 4.1), and *learning meta-policies* (Section 4.2) can be better than $\pi^{\text{RLHF}}$. We observe in Figure 11 that the meta-policy is slightly preferred over CLARIFYFLEX, however this ordering may not be universal: when historical data is not representative of future conversations, we may prefer CLARIFYFLEX over learning a meta-policy.

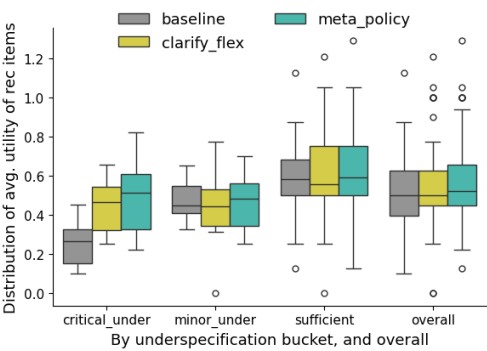

Figure 11: Our learned meta-policy outperforms baseline across all under-specification buckets, especially when queries are critically under-specified. And it converges to baseline when queries are sufficiently specified.

# 5 RELATED WORK

Even though LLMs are powerful conversationalists and recommenders [He et al., 2023], they have many failure modes [Borji, 2023] such as generating hallucinations or failing to complete more complex reasoning tasks [Bubeck et al., 2023] (Section 8). Regarding LLM-powered conversations that require stronger collaboration between two parties, Lin et al. [2023] introduce the concept of "decision-oriented dialogues" and show that current LLMs still are far from human performance. In this paper, we investigate a specific cause (query under-specification) and show how we can improve LLMs for them.

We conjecture that query under-specification is an artifact introduced or amplified during post-training and alignment workflows such as reinforcement learning from human feedback (RLHF) [Ouyang et al., 2022]. In RLHF, LLMs are fine-tuned to output results that align with the preferences of annotators. Status quo approaches focus on pairwise comparisons of single-step responses to a given input query. As such, well-specified and/or simpler queries that admit multiple possible, high-quality responses *without* the need for clarification questions may be over-represented during fine-tuning. Additionally, when annotators *do* encounter under-specified queries, their preferences about how to handle ambiguity may differ in meaningful ways from those of end-users, skewing the learned policy. For example, Singhal et al. [2023] observe that annotators tend to prefer longer responses—which help to "cover all bases" when queries are under-specified—relative to end-users, who must bear the cognitive cost of LLM verbosity. Annotators may also provide feedback they feel is "expected" of them that diverges from their true conversational preferences (due to the Hawthorne effect; see McCambridge et al. [2014]).

Query under-specification has been studied and addressed in information retrieval [Dang and Croft, 2010, Azad and Deepak, 2019]. There are two broad approaches: algorithmic or user-centric techniques. Algorithmic approaches include query expansion [Azad and Deepak, 2019], query reformulation [Dang and Croft, 2010] etc. User-centric approaches focus on asking good clarifications [Rao and Daumé III, 2018, Majumder et al., 2021]. Hybrid approaches are possible: for instance, Diao et al. [2023] use active learning to determine what questions to ask in an LLM's context window so as to improve its reasoning. We take a user-centric approach of seeking clarification, and rely on a suitably prompted LLM (rather than a separate active learning policy) to discover appropriate questions to ask.

We showed that LLMs are misaligned when queries are under-specified. Others have shown misalignment for other reasons (e.g. toxicity [Bai et al., 2022]) and studied better ways to align LLMs. There are two kinds of approaches to align LLMs better: fine-tuning (e.g., DPO [Rafailov et al., 2023], KTO [Ethayarajh et al., 2024], RLHF [Ouyang et al., 2022], etc.) and prompt injection (e.g., Constitutional A, I [Bai et al., 2022], meta-prompting [Qin and Eisner, 2021]). We take the latter approach and extend the meta-prompting of Qin and Eisner [2021] to work not only with soft-prompts but with natural language prompts and black-box LLMs.

Our proposed interventions rely on asking users clarification questions. User studies conducted with search engines [Zamani et al., 2020] and pre-LLM conversation systems [Christakopoulou et al., 2016] demonstrated that users *do* engage with clarifying questions in those contexts. Conducting user studies in LLM-based chatbots to assess users' propensity to answer questions is an exciting avenue for future work.

We frame the conversation between a user and chatbot as a PODP, which is mathematically equivalent to a partially observable Markov Decision Process (POMDP) [Littman, 2009]. Others have framed the interactions as multiple rounds of bandit interactions [Zuo et al., 2022], but as we argued before, single-turn utility maximization is too myopic for multi-turn conversational outcomes. Thus, we adapt solution concepts from POMDP like Q-learning [Watkins and Dayan, 1992], information-gathering [Sadigh et al., 2016] for use with LLM-induced policies.

# 6 LIMITATIONS

While our empirical results demonstrate that both of our proposed interventions improve expected conversation-level utility when queries are under-specified, it is worth noting some limitations associated with the way we have modeled user-chatbot interactions. First, we note that our model relies on the assumption that users are both *willing* and *able* to answer clarification questions when asked—that is, that they will (1) "tolerate" the questions with high probability (i.e., will not defect by exiting the conversation), and (2) truthfully reveal their preferences. In practice, the propensity and ability to answer will vary among users and over query intent domains (e.g., due to personal preferences, epistemic uncertainty regarding a specific topic, etc.).

In our empirical results, the optimistic nature of these assumptions is offset by the conservative nature of the information gain we consider: oftentimes, LLM questions will ask for more granularity about already-revealed $\theta$s, and while real users would often be able to provide such detail, our lossy, parameterized approximation cannot. As such, any improvement in expected reward associated with sequential response strategies that incorporate uncertainty reduction at $t_0$ may be underestimated. We have focused on undiscounted expected utility maximization, but the incorporation of a discount rate would be one way to incorporate heterogeneity with respect to question tolerance. Human validation of our proposed interventions will also be critical: while the interventions are well-motivated from an information-theoretic perspective, for some users, the marginal improvement in

expected utility may not outweigh the cognitive cost associated with having to answer questions.

Additionally, we note that while we have relied on helper LLMs to classify queries and responses (i.e., with respect to under-specification, and response strategy), human validation of these classifiers is an important next step. We have considered a relatively restricted intent domain, but in more general settings, reasonable annotators may disagree about whether a query is under-specified when they do not have access to ground-truth $\theta$. Relatedly, we have focused on a recommendation setting (i.e., movie recs) that admits objective computation of utility; extension of our approach to intents characterized by more subjective evaluation criteria may require alternative approaches to modeling utility.

In the data-based intervention outlined in Section 4.2, we have assumed that historical conversation logs are representative of the user population and joint distribution over users and queries seen in the online setting. This assumption may be violated in practice, with potentially negative consequences for meta-policy performance. Our estimates regarding the prevalence of query underspecification may also contain artifacts—e.g., due to small sample size, and non-stationarity of the user population.

Finally, we have made assumptions regarding the prompt-based steerability of LLMs, along with the ability of LLMs to select "good" clarification questions when prompted to clarify. Empirical validation of these assumptions on a broad set of LLMs, along with studying the generation and selection of marginal information-gain maximizing questions, are important directions for future work.

# 7 CONCLUSION

This paper explores how user underspecification affects the behavior of LLM-based chatbots that are fine-tuned with human feedback. We show that chatbots have difficulty handling vague user requests and explain how this issue stems from the annotation process of LLMs. Our study of a public chat logs dataset confirms that this problem is common – over 25% of the queries are highly underspecified. We formulate the problem of underspecification as a partially observable decision process (PODP) and generate synthetic data from a recommendation scenario with hidden item values for experimental evaluation. Our experiments show that pre-trained LLMs perform poorly on underspecified user queries and propose a method to adjust LLMs through prompting (with learned control messages). We demonstrate that our lightweight learning method can effectively leverage previous conversation data to improve the response behavior of LLM-based chatbots for recommendation tasks.

## Acknowledgements

We thank Allen Nie, Tara Safavi, Jay Stokes, John Dickerson, Philip Resnik for insightful discussions and the anonymous reviewers of UAI'24 for their feedback.

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

# On Overcoming Miscalibrated Conversational Priors in LLM-based Chatbots (Supplementary Material)

Christine Herlihy[1]     Jennifer Neville[2]     Tobias Schnabel[2]     Adith Swaminathan[2]

[1]Department of Computer Science, University of Maryland, College Park, MD USA
[2]Microsoft Research, Redmond, WA USA

## A   HELPER LLM-BASED CLASSIFIERS

In this section, we provide descriptions, system messages, and validation results for each of the helper-LLM-based classifiers that we rely on throughout the paper.

### A.1   LLM-BASED QUERY UNDERSPECIFICATION CLASSIFIER

**Task description:** We use a helper LLM to map queries from our synthetic and real-world corpora to a set of class labels that describe the extent to which a given query is (or is not) under-specified, i.e., {CRITICAL UNDER, MINOR UNDER, SUFFICIENT}. We introduce these labels in Section 3.1, where we discuss them within the context of the labels we (i.e., human annotators) manually assign to a randomly sampled subset of the OpenAssistant corpus, but they also apply when the helper LLM is asked to classify queries. For convenience, we repeat them below:

- CRITICAL UNDER: One or more important factors upon which an answer to this query might depend are not specified or are unknown; (annotators agree that) it is difficult to provide a high-quality response without knowing these factors.

- MINOR UNDER: Less important factors that the query might depend on are not specified or are unknown; however, it is possible to provide a high-quality response even without knowing these factors.

- SUFFICIENT: All important factors upon which an answer to this query might depend are sufficiently specified.

**Helper LLM prompt**   The prompt that we provide to the helper LLM for this task is shown below; it is also included within the `helper_task_system_messages.json` file contained within our supplemental materials.

```
{
    "classify_queries_multiclass": "For each query in this list <list>{{input.question}}</
        list>, assign exactly one of the following labels:\n
        - sufficient: All important factors upon which an answer to this query might
            depend are sufficiently specified.\n
        - minor_under: One or more less important factors upon which an answer to this
            query might depend are not specified or are unknown; however, it is possible
            to provide a high-quality response even without knowing these factors.\n
        - critical_under: One or more important factors upon which an answer to this
            query might depend are not specified or are unknown; it is difficult to
            provide a high-quality response without knowing these factors.\n
    You MUST assign EXACTLY ONE label from the list above.\n
    Return your answer as a string.\n
    DO NOT answer any questions contained in the query, or include any expository text.\n
    The result should be DIRECTLY parsable in Python."
}
```

**Helper LLM configuration:**  We use GPT-4 [Achiam et al., 2023] for all query underspecification classification calls.

**Validation:**  We use our synthetic query corpus to validate our use of this LLM-based underspecification classifier. As we describe in Section 3.2.1 and detail in Appendix C.1, by virtue of how we construct these queries, we control the number of attributes that are revealed. As such, we have access to ground-truth underspecification labels defined in terms of the number of revealed attributes, referred to (with slight abuse of notation) as $|q|$ in the mapping shown below. Note that $|q|$ takes values in $\{0, \ldots, |\theta| - 1\}$ for masked queries, and will be equal to $|\theta|$ for sufficiently specified queries, where $|\theta|$ refers to the cardinality of the intent-specific attribute space.

$$q \mapsto \begin{cases} \text{CRITICAL UNDER} & |q| \le 1, \\ \text{SUFFICIENT} & |q| = |\theta|, \\ \text{MINOR UNDER} & \text{otherwise.} \end{cases}$$

We evaluate our LLM-based query underspecification classifier on our synthetic query corpus, which contains 600 queries split across the following intent domains: movie recommendation, gift recommendation, and plant recommendation. We report performance metrics and confusion matrices over all synthetic queries, and broken down by intent-specific queries below.

|  | precision | recall | f1-score | support |
|---|---|---|---|---|
| critical_under | 0.583 | 0.139 | 0.224 | 202 |
| minor_under | 0.443 | 0.472 | 0.457 | 398 |
| sufficient | 0.720 | 0.873 | 0.789 | 600 |
| accuracy |  |  | 0.617 | 1200 |
| macro avg | 0.582 | 0.495 | 0.490 | 1200 |
| weighted avg | 0.605 | 0.617 | 0.584 | 1200 |

Table 4: Classifier performance: over all intents

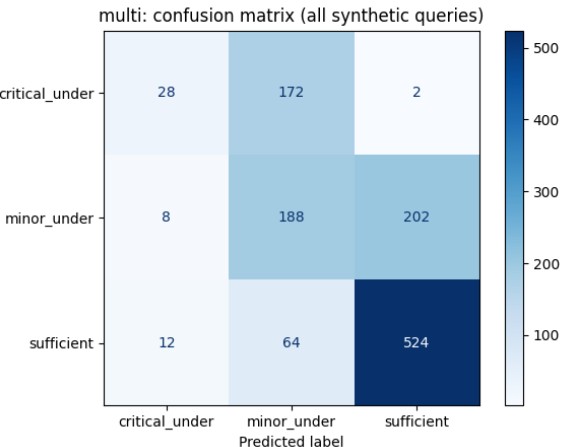

Figure 12: Confusion matrix: all intents

|              | precision | recall | f1-score | support |
| ------------ | --------- | ------ | -------- | ------- |
| critical_under | 0.059   | 0.010  | 0.017    | 99      |
| minor_under    | 0.284   | 0.214  | 0.244    | 196     |
| sufficient     | 0.642   | 0.925  | 0.758    | 295     |
| accuracy       |         |        | 0.536    | 590     |
| macro avg      | 0.328   | 0.383  | 0.340    | 590     |
| weighted avg   | 0.425   | 0.536  | 0.463    | 590     |

Table 5: Classifier performance: movie recommendation queries

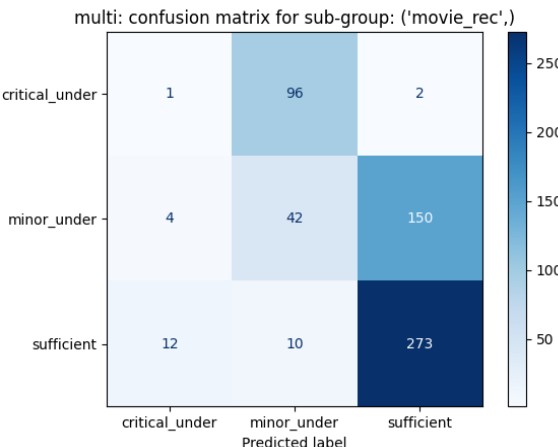

Figure 13: Confusion matrix: movie recommendation queries

|              | precision | recall | f1-score | support |
| ------------ | --------- | ------ | -------- | ------- |
| critical_under | 0.852   | 0.371  | 0.517    | 62      |
| minor_under    | 0.652   | 0.861  | 0.742    | 122     |
| sufficient     | 0.928   | 0.908  | 0.918    | 184     |
| accuracy       |         |        | 0.802    | 368     |
| macro avg      | 0.811   | 0.713  | 0.725    | 368     |
| weighted avg   | 0.824   | 0.802  | 0.792    | 368     |

Table 6: Classifier performance: gift recommendation queries

|              | precision | recall | f1-score | support |
| ------------ | --------- | ------ | -------- | ------- |
| critical_under | 1.000   | 0.098  | 0.178    | 41      |
| minor_under    | 0.357   | 0.512  | 0.421    | 80      |
| sufficient     | 0.683   | 0.694  | 0.689    | 121     |
| accuracy       |         |        | 0.533    | 242     |
| macro avg      | 0.680   | 0.435  | 0.429    | 242     |
| weighted avg   | 0.629   | 0.533  | 0.513    | 242     |

Table 7: Classifier performance: plant recommendation queries

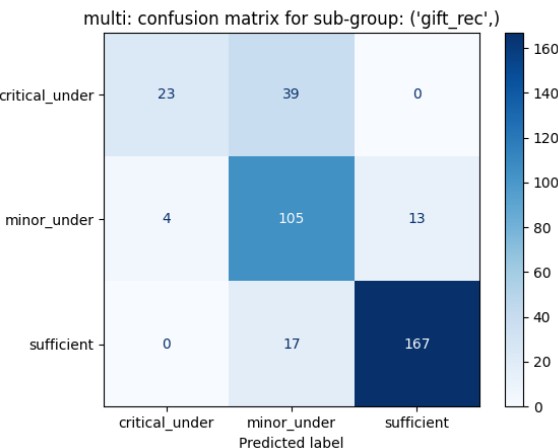

Figure 14: Confusion matrix: gift recommendation queries

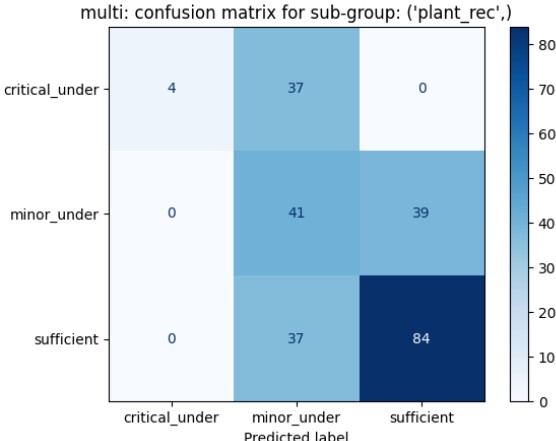

Figure 15: Confusion matrix: plant recommendation queries

## A.2 LLM-BASED RESPONSE STRATEGY CLASSIFIER

**Task description:** We use a helper-LLM-based $\tau$ classifier to map chatbot natural language responses to a set of labels intended to characterize a given response's syntactic and semantic contents. We primarily use this classifier as a way of assessing whether and to what extent the behaviors we seek to induce via modified system messages *actually* produce observable effects in the intended direction(s) and/or converge with the behavior of $\pi^{\mathrm{RLHF}}$.

The label set we use for this classifier includes the set of response strategies that we refer to as $\mathcal{T}$ throughout the paper—i.e., {INTERROGATE, CLARIFY, HEDGE},and also includes additional options—i.e., {DIRECT RESPONSE, REFUSE, MISCELLANEOUS, MISSING}. While we do not explicitly induce this latter set of behaviors, we need the DIRECT RESPONSE option to characterize the baseline system behavior and (more broadly) uncertainty-agnostic LLM responses in general. The REFUSE, MISCELLANEOUS, and MISSING options are needed to characterize the behavior of $\pi^{\mathrm{RLHF}}$ in open-domain settings such as the OpenAssistant corpus we consider, as well as to handle rare parsing/extraction errors that result in inadvertently blank LLM responses. The defining characteristics of each response strategy are presented/contained within the task system message in the next section.

**Helper LLM prompt:** The prompt that we provide to the helper LLM for this task is shown below; it is also included within the `helper_task_system_messages.json` file contained within our supplemental materials.

```
{
    "sm_map_llmr_to_tau": str = "For each (query,response) in this list <list>{{input.pair
        }}</list>, map the response to exactly one of the following labels:\n

        - interrogate: The response contains a large number (i.e., more than 3) of follow-
            up questions and and does NOT contain plausible responses conditioned on
            possible answers to these questions.\n
        - clarify: The response contains a limited number (i.e., 3 or less) of follow-up
            questions and does NOT contain plausible responses conditioned on possible
            answers to these questions.\n
        - hedging: The response does not commit to one specific answer but instead
            provides many plausible/possible/qualified answers, options, or conditions
            under which certain answers/options may or may not hold. It may also discuss (
            potentially conflicting) different view points without taking a definitive
            stance.\n
        - direct_response: The response does NOT contain questions. The response does NOT
            contain multiple plausible answers, with corresponding descriptions of
            conditions or criteria under which each response would be suitable.\n
        - refuse: The response contains an explicit or implicit refusal to answer. It may
            mention criteria which would be needed in order to provide an answer, but it
            does NOT contain plausible responses conditioned on these criteria.\n
        - misc: The response may describe, summarize, or try to explain the query, or
            appear to follow instructions provided in the query (rather than answer an
            information-seeking request or ask clarifying questions).\n
        - missing_response: The response is empty or blank.\n

    You MUST assign exactly one label from the list above.\n
    Return your answer as a string.\n
    DO NOT answer any questions contained in the response, or include any expository text
        .\n
    The result should be DIRECTLY parsable in Python."
}
```

**Helper LLM configuration:** We use GPT-4 [Achiam et al., 2023] for all query underspecification classification calls.

**Validation:** We manually annotate $\pi^{\text{RLHF}}$ responses to a subset of the OpenAssistant corpus that we consider, and use these human-annotator assigned ground-truth $\tau$s to validate our helper LLM-based $\tau$ classifier. We note that some of our $\tau$s of interest are not sufficiently represented amongst the $\pi^{\text{RLHF}}$ responses (i.e., CLARIFY, HEDGE, INTERROGATE). We thus use our system-message-based interventions to induce responses for these strategies and include them (unlabeled) in our manually annotated subset. We report classification performance metrics and a confusion matrix below.

|                 | precision | recall | f1-score | support |
|-----------------|-----------|--------|----------|---------|
| clarify         | 0.763     | 0.935  | 0.841    | 31      |
| direct_response | 0.822     | 0.903  | 0.861    | 154     |
| hedging         | 0.925     | 0.649  | 0.763    | 57      |
| interrogate     | 1.000     | 0.680  | 0.810    | 25      |
| misc            | 0.154     | 0.250  | 0.190    | 8       |
| refuse          | 0.667     | 0.400  | 0.500    | 5       |
| accuracy        |           |        | 0.807    | 280     |
| macro avg       | 0.722     | 0.636  | 0.661    | 280     |
| weighted avg    | 0.831     | 0.807  | 0.808    | 280     |

Table 8: $\tau$-classifier performance on human-annotated LLM responses to OpenAssistant queries

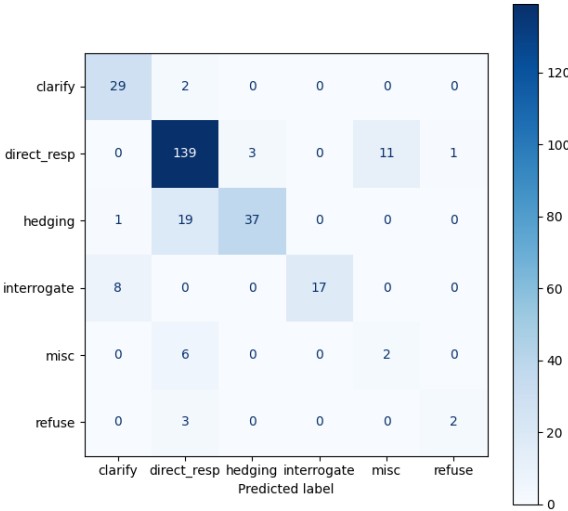

Figure 16: Confusion matrix: annotated OpenAssistant query responses

# B    RESPONSE-STRATEGY SYSTEM MESSAGES

In this section, we report the system messages used to operationalize each response strategy $\tau \in \mathcal{T}$ that we consider in the main paper, along with the data-agnostic interventions discussed in Section 4.1.

```
{
 "response strategies":
 {
     "baseline": "None",
     "interrogate": "When you receive a query, always interrogate the user about all
          factors upon which the answer might depend---but that have not been specified---
          so that you will be able to produce a good answer.",
     "clarify": "When you receive a query, always ask the user about up to 3 most
          relevant factors upon which the answer might depend---but that have not been
          specified---so that you will be able to produce a good answer.",
     "hedge": "When you receive a query, always identify important factors upon which the
           answer might depend---but that have not been specified---and then provide a
          plausible response conditioned on each of these factors.",
 }
 "data-agnostic interventions":
 {
   "CoT": "When you receive a query, ask yourself whether you have sufficient information
        to provide a good answer, and then respond accordingly.",
   "clarify_flex": "When you receive a query, if the query depends on a set of important
       factors that have not been specified, ask the user about the most relevant factors
        that have not been specified so that you will be able to produce a good answer;
       otherwise, respond directly."
 }
}
```

# C MOTIVATING EXPERIMENTS

## C.1 SYNTHETIC DATASET CONSTRUCTION

We follow the synthetic query construction process outlined in Section 3.2.1 and formalized in Algorithm 1, along with the parameterized templates shown in Appendix C.2 to generate a corpus of sufficiently specified queries. Then, for each sufficiently specified query, we approximate a *partially specified* version by randomly selecting the number of attributes to omit, $n \in \{1, \ldots, |\Theta_i|\}$. We keep the intent-declaring first sentence unchanged, shuffle the remaining attribute-sentences, draw a subset of sentences to *omit*—i.e., with cardinality $n$— and concatenate the remaining sentences.

---

**Algorithm 1** GENERATE SYNTHETIC QUERY

---

1: **function** GenQuery($\mathcal{I}, \Theta_{(.)}$)
2:   $i \sim \mathcal{I}$ {Draw intent}
3:   **for** $\theta \in \Theta_i$ **do**
4:     selected option(s) $\leftarrow X' \sim X_\theta$
5:     template$_i \leftarrow$ template$_i \cup$ selected option(s)
6:   **end for**
7:   $q_s \leftarrow$ template$_i$ {Sufficient query := filled-in template}
8:   $n \sim U(\{1, \ldots, |\Theta_i|\})$ {Draw # of attrs to mask}
9:   $\Theta_i^m \sim U(S_n(\Theta_i))$ {Draw $n$ masked attrs}
10:  $\Theta_i^r := \Theta_i \setminus \Theta_i^m$ {Determine revealed attrs}
11:  $q_m \leftarrow$ concat($\Theta_i^r$) {Build masked version of $q_s$}
12:  **return** $q_s, q_m$ {Return sufficient & masked queries}

---

## C.2 SYNTHETIC QUERY TEMPLATES AND PARAMETER OPTIONS

We use the intent-specific templates and parameter category-option mappings shown below in conjunction with the query construction procedure outlined in Appendix C.1 to generate our synthetic queries:

**Intent-specific query templates:**

```
{
    "movie_rec": {
      "value": "I am looking for a movie recommendation.\n
      The genre should be \"{param_0}\".\n
      It should have been released \"{param_1}\".\n
      The intended audience includes \"{param_2}\".\n
      The runtime should be \"{param_3}\".\n
      Please provide movie recommendations that satisfy all of my requirements."
    },

    "gift_rec": {
      "value": "I am looking for a gift recommendation.\n
      The recipient likes \"{param_0}\".\n
      The recipient is \"{param_1}\" years old.\n
      The recipient prefers gifts to be \"{param_2}\" in nature.\n
      My budget to purchase the recipient a gift is in the \"{param_3}\" range.\n
      Please provide gift recommendations that satisfy all of my requirements."
    },

      "plant_rec": {
      "value": "I am looking for a house plant recommendation.\n
      I prefer a plant that \"{param_0}\".\n
      I'm willing to expend a \"{param_1}\" amount of effort to care for the plant.\n
      My house gets a \"{param_2}\" amount of natural light.\n
      I live \"{param_3}\".\n
      Please provide house plant recommendations that satisfy all of my requirements."
```

```
27        }
28 }
```

**Intent-specific query parameter category-option mappings:**

```
1  {
2        "movie_rec": {
3          "param_0": {"cat": "genre",
4                      "opts": ["Action", "Adventure", "Animation", "Biography", "Comedy", "
                           Crime", "Documentary", "Drama", "Fantasy", "Film Noir", "History",
                           "Horror", "Musical", "Mystery", "Romance", "Sci-Fi", "Sport", "
                           Superhero", "Thriller", "War", "Western"],
5                      "max_sel_allowed": 2,
6                      "pref_constraint_type": "set_valued"},
7          "param_1": {"cat": "release date",
8                      "opts": ["in the 1980s", "in the 1990s", "in the 2000s", "in the past
                           few years"],
9                      "max_sel_allowed":1,
10                     "pref_constraint_type": "numeric_range"},
11         "param_2": {"cat": "who will be watching",
12                     "opts": ["children", "adults and children", "teenagers younger than 17
                           ", "adults only"],
13                     "max_sel_allowed": 1,
14                     "pref_constraint_type": "set_valued"},
15         "param_3": {"cat":  "runtime",
16                     "opts": ["less than 90 minutes", "90-104 minutes", "105-119 minutes",
                           "120 minutes or more"],
17                     "max_sel_allowed": 1,
18                     "pref_constraint_type": "set_valued"}
19       },
20
21       "gift_rec": {
22         "param_0": {"cat": "recipient interests",
23                     "opts": ["outdoors", "crafts", "technology", "books", "active play/
                           sports/fitness", "food/cooking", "music and arts", "apparel/
                           fashion/style"],
24                     "max_sel_allowed": 2,
25                     "pref_constraint_type": "set_valued"},
26         "param_1": {"cat": "recipient age range",
27                     "opts": ["3-5", "6-12", "13-17", "18-40", "41-60", "61+"],
28                     "max_sel_allowed": 1,
29                     "pref_constraint_type": "numeric_range"},
30         "param_2": {"cat": "recipient preferred gift type",
31                     "opts": ["practical and everyday", "personalized and sentimental", "
                           adventurous and experience-driven", "luxurious and pampering", "
                           high-tech and innovative", "creative and artistic", "sustainable
                           and eco-friendly"],
32                     "max_sel_allowed": 1,
33                     "pref_constraint_type": "set_valued"},
34         "param_3": {"cat": "giver budget",
35                     "opts": ["less than $20", "$20-49", "$50-99", "$100-199", "$200+"],
36                     "max_sel_allowed": 1,
37                     "pref_constraint_type": "numeric_range"}
38       },
39
40       "plant_rec": {
41         "param_0": {"cat": "plant physical attributes",
42                     "opts": ["has light green leaves", "has dark green leaves", "has multi
                           -colored leaves", "yields edible produce", "produces a single
                           bloom", "produces multiple blooms", "is short in stature", "is
                           tall in stature"],
```

```
43                      "max_sel_allowed":  2,
44                      "pref_constraint_type": "set_valued"},
45          "param_1": {"cat": "level of effort",
46                      "opts":  ["low", "moderate", "high"],
47                      "max_sel_allowed":  1,
48                      "pref_constraint_type": "set_valued"},
49          "param_2": {"cat": "amount of natural light",
50                      "opts":  ["low", "indirect", "moderate", "bright"],
51                      "max_sel_allowed":  1,
52                      "pref_constraint_type": "set_valued"},
53          "param_3": {"cat": "household members",
54                      "opts":  ["by myself", "with young children", "with pets", "with a
                           roommate with seasonal allergies"],
55                      "max_sel_allowed": 1,
56                      "pref_constraint_type": "set_valued"}
57      }
58 }
```

## C.3 STABLE DIFFUSION EXAMPLE: POLICY-INDUCED RESPONSES

Here, we present a query from the OpenAssistant dataset [Köpf et al., 2023] as an example of a situation where CLARIFY may be preferred over DIRECT RESPONSE and HEDGE. Table 9 presents the policy-induced responses generated by $\pi^{\text{RLHF}}$, HEDGE, and CLARIFY.

**Query text:** "Please give me a prompt for stable diffusion to generate a good-looking image."

| Response strategy $\tau \in \mathcal{T}$ | Response text |
| --- | --- |
| baseline | "Create an image that visually represents the process of stable diffusion, showcasing the gradual and even spread of particles throughout a medium. Use a color gradient to highlight the movement and distribution of particles, and ensure the overall aesthetic is pleasing and balanced." |
| hedge | The query is asking for a prompt to generate a good-looking image using stable diffusion. However, the important factors that are not specified include: 1. The type of image or subject matter desired (e.g., landscape, portrait, abstract, etc.) 2. The style or aesthetic the user prefers (e.g., realistic, surreal, impressionistic, etc.) 3. The color scheme or palette the user prefers (e.g., warm, cool, monochromatic, etc.) Given these factors, here are some plausible responses: 1. If the user wants a landscape image in a realistic style with a warm color scheme, a possible prompt could be: "Generate a sunset over a serene lake with autumn foliage using stable diffusion." 2. If the user prefers a portrait in a surreal style with a cool color scheme, a possible prompt could be: "Generate a portrait of a woman with flowing blue hair and eyes that mirror the ocean using stable diffusion." 3. If the user is interested in an abstract image in a monochromatic color scheme, a possible prompt could be: "Generate an abstract image using stable diffusion that represents the concept of tranquility in shades of blue." |
| clarify | 1. What type of image are you looking to create? (e.g., abstract, realistic, geometric) 2. Are there any specific colors or color schemes you would like to incorporate into the image? 3. Do you have any specific themes or elements you want to be included in the image? (e.g., nature, space, patterns) |

Table 9: Policy-induced responses to the stable diffusion example query.

## C.4 SIMULATING USER RESPONSES TO LLM QUESTIONS

In Section 3.2.3, we discuss how we leverage a series of helper LLM calls to construct templatized responses to LLM questions based on the overlap (or lack thereof) between the questions and the user's true preferences, $\theta$. Here we provide the system messages used in these helper calls. The first system message is for the extraction of recommended items and/or questions from LLM responses. The second system message helps us to construct a mapping from the LLM's questions to the user's true preferences, $\theta$, such that we can determine what subset of previously masked attributes can be "revealed" in the templatized user response we construct.

```
{"extract_recs_and_questions":

    "For each response in this list <list>{{input."""  + f'{field_to_use}' +"""}}</list>,
        read the response carefully and:\n

            1. Extract the titles of each and every movie recommendation that appears;
                they may show up as a list of titles.\n
                Do not extract any additional metadata but DO extract any mentioned titles
                    ; represent each title as a string.\n

            Format your answer for task 1 as shown below:\n
```

```
10              ["rec" for rec in recommended movies] OR [], ONLY if NO movie
                    recommendations appear.
11
12          2. Extract any questions that appear; they may be prefaced with a request to
                specify preferences, and/or show up as a list of questions.\n
13              DO extract ANY mentioned questions; represent each question as a string.
14
15              Format your answer for task 2 as shown below:\n
16
17              ["question" for question in questions] OR [], ONLY if NO questions appear
                    .\n
18
19          Return your results as a dict:\n
20
21              {"recs": [response to task 1], "questions": [response to task 2]} \n
22
23          DO NOT answer any questions contained in the response, or include ANY
                expository text.\n
24          The result should be DIRECTLY parsable as a valid dict in Python.""""
25 }
```

```
1  {
2      "map_questions_to_thetas":
3
4          "You will receive a list containing sets of questions.\n
5          Each question is issued by an assistant to a user, in response to a movie
                recommendation request submitted by the user.\n
6          For each set of questions in this list <list>{{input.questions}}</list>,\n
7          You have the ability to ask the user about their preferences for each of the
                following movie attributes:
8              [genre, release date, who will be watching, runtime].\n
9          Note that 'who will be watching' is related to the user's preferences for the
                movie's rating.
10         For each set of questions, map each question to one of these attributes IF asking
                about this specific attribute would allow you to answer the question.\n
11         If none of the attributes would give you the information you need to answer a
                given question, map that question to "None".\n
12         Format your response as a list of strings, as shown in the example below:\n
13             ['genre', 'release date', 'None'] \n
14         DO NOT answer any questions contained in the response, or include any expository
                text.\n
15         The result should be DIRECTLY parsable as a list of strings in Python."
16 }
```