# OpenReview forum: "On Overcoming Miscalibrated Conversational Priors in LLM-based ChatBots"
_auai.org/UAI/2024/Conference — UAI 2024 poster_

### Official Review · Reviewer_np8F · 2024-03-19

**Q2-1 Originality-Novelty:** 3
**Q2-2 Correctness-Technical Quality:** 2
**Q2-5 Clarity Of Writing:** 3

**Q1 Summary And Contributions:**

The paper addresses the challenges faced by Large Language Model (LLM)-based chatbots, particularly when dealing with under-specified queries from users. It identifies a key issue where chatbots often respond poorly to these queries by making incorrect assumptions, providing lengthy and hedging responses, or refusing to answer. This issue is attributed to miscalibrated conversational priors resulting from the LLM fine-tuning process, which may not adequately capture multi-turn conversation utility or represent the diverse preferences of users outside the annotation process.

**Q2-3 Extent To Which Claims Are Supported By Evidence:**

3: Good: the main claims are supported by convincing evidence (in the form of adequate experimental evaluation, proofs, (pseudo-)code, references, assumptions).

**Q2-4 Reproducibility:**

2: Fair: key resources (e.g. proofs, code, data) are unavailable but key details (e.g. proof sketches, experimental setup) are sufficiently well-described for an expert to confidently reproduce the main results.

**Q3 Main Strengths:**

The paper is significant: the paper targets the prevalent issue of under-specified queries in conversational AI, a significant challenge in real-world applications of LLM-based chatbots. This focus is particularly relevant given the increasing reliance on chatbots for a wide range of services, making the work highly impactful.
The empirical results are fine - the authors show that the proposed methods effectively improve the conversational utility of chatbots in handling under-specified queries. This demonstrates the potential of the approaches to enhance user satisfaction and engagement with chatbots.

**Q4 Main Weakness:**

The lightweight learning approach relies heavily on the availability and quality of logged conversation data. In cases where such data is sparse, noisy, or unrepresentative of the broader user base, the effectiveness of the re-calibration process could be compromised. Theorefore, it's quite important to release those data when the draft is out so users can actually access them and reproduce results.

The goal of this paper, according the the abstract and intro, is addressing under-specification queries from users. This itself might be important and realistic, but this seems to have limited relationship with the title "Conversational Priors".

As pointed out in the paper, in a PODP, it is critical to balance information-seeking (exploration) against utility-maximizing (exploitation). But the paper didn't suggest a practical solution. Selecting prompts dynamically during a conversation but is it feasible to do for platforms like chatbot arena and production models like Claude (assuming the companies might collect enough user data)? Also the trade-off would be interesting itself.

**Q5 Detailed Comments To The Authors:**

It would be valuable to include a deeper analysis of user engagement and behavior in response to clarifying questions. Conducting user studies to evaluate the willingness of users to interact with chatbots seeking clarification and their satisfaction with the interaction outcomes could provide a more nuanced understanding of the effectiveness of these strategies. This aspect is important in analyzing the actual social impact of the proposed method [1]


[1] To Trust or to Think: Cognitive Forcing Functions Can Reduce Overreliance on AI in AI-assisted Decision-making

The authors should also discuss relation with recent works showing the miscalibration and utility-based decision theory of LMs [2, 3]. Esepcially can try to avoid using vague terms like beliefs that are hard to measure but resort to more well-grounded notions like calibration [3]

[2] DeLLMa: A Framework for Decision Making Under Uncertainty with Large Language Models

[3] A Study on the Calibration of In-context Learning

**Q9 Complying With Reviewing Instructions:**

Yes

---

> ### Author Rebuttal · Authors · 2024-04-09
>
> Thank you for your helpful feedback. Please see our responses below:
>
> ### **To address the weaknesses you raise**:
> 1. **Assumptions regarding offline data**:
>     - We agree that high-quality data is required for the offline RL approach to work well. Rather than make strong data assumptions to argue for it, we instead propose a *data-agnostic* approach (Section 4.1).
>     - We believe that designing good prompts (Section 4.1) and learning meta-policies (Section 4.2) can *both* be effective fixes; choosing between these options may depend on the data regime.
>         - When historical data is not representative of future conversations, we may prefer Clarify-Flex over learning a meta-policy.
>         - Alternatively, if users engage exclusively in single-turn interactions, the baseline LLM responses may already be well-calibrated.
>     - We will acknowledge our assumptions about data availability and include a discussion about the relative merits of our two proposed approaches in Section 4.
> 2. **Relationship between underspecification and conversational priors**:
>     - **Characterization of LLM's conversational priors**: We introduce a response taxonomy that can be used to characterize an LLM's response behavior (see Figure 5 for an example featuring GPT-4). We refer to the resulting distribution over response strategies as the "conversational prior" of the LLM-based PODP policy (recall: this policy, $\pi$, maps from conversational prefixes to natural language responses).
>     - **Relationship to query under-specification**: Analyzing conversational priors across different query segments is what allowed us to detect the mishandling of under-specified queries by state-of-the-art LLMs.
> 3. **Feasibility of our proposed approaches**:
>    - **Our approaches**: We propose and evaluate two practical approaches (i.e., Sections 4.1 and 4.2) to address situations where an LLM may be responding sub-optimally to under-specified queries.
>    - **Feasibility**: Both of the approaches we propose rely *only* on API access to blackbox LLMs; choosing between the two may depend on the data regime:
>        - If logged data *is* available, the meta-policy learned in Section 4.2 (where prompts are selected from a small, discrete set) is a tractable alternative to more resource-intensive options such as fine-tuning the underlying LLM on collected user conversations.
>        - Conversely, if we do *not* have access to sufficient, high-quality data, we may prefer the data-agnostic approach of Section 4.1.
>     - **Chatbot Arena; Claude**:
>         - For platforms like Chatbot Arena or instruction-following LLMs like Claude where you are not able to provide a prompt and query/transcript as separate fields, the meta-policy-based prompting approach we propose can be implemented as follows: (1) keep track of the conversational transcript as it progresses (i.e. concatenate user and chatbot responses); (2) at each conversational turn, query the learned meta-policy using the transcript to select the next prompt; (3) concatenate the selected prompt and transcript and pass the resulting text as input (i.e., to the platform or LLM); (4) concatenate the resulting LLM response to the transcript, and continue iteratively.
>
> ### **In response to your suggestions**:
> 1. **Assumptions related to user behavior**:
>     - We agree that in real-world settings, the users' willingness to answer clarifications may impact the realized gains associated with our proposed approaches.
>    - User studies have been conducted with search engines ("Analyzing and Learning from User Interactions for Search Clarification", [Zamani et al. SIGIR'20](https://arxiv.org/abs/2006.00166) ) and pre-LLM conversation systems ("Towards Conversational Recommender Systems", [Christokopoulou et al. KDD'16](https://dl.acm.org/doi/10.1145/2939672.2939746) ) that demonstrated that users *do* engage with clarifying questions in those contexts.
>    - In our work, we diagnose query under-specification as a root cause of some sub-optimal LLM responses, and propose two intervention strategies. User studies of our approaches---and the viability of encouraging clarification in LLM chatbots in general---is an exciting direction for future work!
> 2. **Suggested related works**:
>    - Thank you for referencing recent work on LLMs being used for decision-making problems ( [Liu et al., 2024](https://arxiv.org/abs/2402.02392)) and studying their calibration of in-context learning ( [Zhang et al., 2024](https://arxiv.org/abs/2312.04021) ). However, we are modeling the *user-LLM interaction* as a PODP --- this is very different and unrelated to a user using an LLM to assist in solving some decision problem.
> 3. **Use of the term "belief"**:
>     - We use the word "belief" in the technical rather than colloquial sense of the word---i.e., as is common in the POMDP (Partially Observed Markov Decision Process) literature. We will revise the text to make our semantic intent more clear.

---

### Official Review · Reviewer_Ak9g · 2024-03-21

**Q2-1 Originality-Novelty:** 2
**Q2-2 Correctness-Technical Quality:** 2
**Q2-5 Clarity Of Writing:** 2

**Q10 Ethical Concerns:**

Not applicable.

**Q1 Summary And Contributions:**

This work studies chatbot interaction with a Large Language Model (LLM) that powers a lot of current recommender systems. The key question this paper tackles is that many of these chatbots suffer from under-specified requests. An under-specified request can be classified as a user interaction where the user intent is not clear from the limited or vague prompt given as input to the LLM. An LLM generally responds to these under-specified requests by wrong answers, or long answers which are incompatible with the requests. This work systematically points out that such responses are a result of miscalibration resulting from fine-tuning on small (single-turn) chats annotated by humans which may not capture the full extent of the request.

Then they formulate the chatbot interaction as a Partially Observed Decision Process (PODP) as the annotators may not reliably infer user preferences or intents from their queries. I like the classification of possible responses into six categories in Figure 1, DIRECT RESPONSE, REFUSE, HEDGE, CLARIFY, INTERROGATE, MISSING, MISCELLANEOUS. In fact they show that both for real-world and synthetic queries, even the current SoTA LLM, GPT-4, tends to hedge by producing a lengthy response covering many plausible answers, rather than clarify via a short question. They show that pre-trained LLMs can be sub-optimal for PODPs and derive better policies that clarify under-specified queries when appropriate. Then, they re-calibrate LLMs by prompting them with learned control messages to approximate the improved policy. Finally, they show that their policies of using control messages recalibrate the LLM in several recommendation tasks.

**Q2-3 Extent To Which Claims Are Supported By Evidence:**

3: Good: the main claims are supported by convincing evidence (in the form of adequate experimental evaluation, proofs, (pseudo-)code, references, assumptions).

**Q2-4 Reproducibility:**

3: Good: key resources (e.g. proofs, code, data) are available and key details (e.g. proofs, experimental setup) are sufficiently well-described for competent researchers to confidently reproduce the main results.

**Q3 Main Strengths:**

1) They show from analyzing publicly available chat logs that query under-specification is common. This is a strong motivation behind this work.
2) Formulating the problem as a PODP process makes sense. Since the annotators may not fully recover the user intent, and the user intent may shift following long interactions, this formulation lays down a strong foundation for future analysis.
3) I like the categorization of the possible response of the LLM into the six types, and the lightweight solution of asking to clarify when appropriate.

**Q4 Main Weakness:**

1) The section 2.1 on introducing and formulating the PODP needs to be explained in more detail. See questions.
2) It is not exactly clear to me how the context-aware clarification is done. See questions.
3) Some experimental details are missing and need to be explained in more detail. See questions.

**Q5 Detailed Comments To The Authors:**

1) It is not clear to me how the LLM can answer context-aware clarification as discussed in section 4.1 data agnostic intervention. Simply asking clarification questions may not be sufficient to grasp the latent user intent or even when asking more clarification based on the context is not clear. Can you elaborate on this?

2) In section 4.2 data based intervention what is this meta policy $\beta$? Is it a policy that takes into account all relevant intervention strategies based on the context? Moreover, the proposed solution as pointed out requires human intervention on a lengthy chat which may be infeasible and sample inefficient. Also how to update the policy weights of $\beta$, using PPO?

3) In the same context, the authors argue an off-policy-based approach where they have access to offline data and use it to possibly devise a data-informed intervention. First, I think this assumption of the existence of offline data and knowledge base should be formally stated. Secondly, this also requires some coverage assumption which must ensure that sufficient long and informative chat history corresponding to the context of the chat interaction actually exists in the dataset. The authors should clarify more on this.

4) Similarly in the same section they propose an alternative Q-learning strategy to train a regressor to map the context and the prompt to some real number. It is not clear to me what is the reward in this setting. This goes back to the discussion of the PODP framework. This partially observed framework seems similar to the POMDP setting. However, the formulation of rewards in PODP is not clear to me (also for the Q-learning approach). Similarly how they intend to use a Q-learning approach for a partially observed setting is not clear to me. This requires more clarification.

**Q9 Complying With Reviewing Instructions:**

Yes

---

> ### Author Rebuttal · Authors · 2024-04-09
>
> Thank you for your helpful feedback. Please see our responses below:
>
> ### **To address the weaknesses you raise**:
> 1. **PODP formulation**:
>    - We apologize for the confusion and take this opportunity to clarify. In Section 2, we frame user-chatbot interactions as a partially observable decision process (PODP), and in Section 4.2, we consider a *separate* decision-making task---i.e., that of learning a meta-controller using logged interaction data.
>    - We answer PODP questions around the use of offline data and the reward function in our responses below.
>    - We will revise notation to avoid inadvertent collisions and provide more detail when describing these two decision-making problems.
> 2. **Context-aware clarification; experimental details**:
>    - Please see our responses to your specific questions below.
>
> ### **To answer your questions**:
> 1. **Context-aware clarification**:
>     - As you point out, we are only able to benefit from asking clarification questions if/when the user *responds* with relevant information that allows the chatbot to better infer the user's *true* intent and construct a more tailored response.
>         - User studies in search engines (``Analyzing and Learning from User Interactions for Search Clarification'', [Zamani et al. SIGIR'20](https://arxiv.org/abs/2006.00166)) demonstrate that user responses to (some forms of) clarifications *do* lead to better results in that context.
>    - In Section 4.1, we task the LLM to create "good" clarification questions without explicitly specifying *which* questions to ask.
>         - Studying how and when LLMs can ask "good" clarifications that users engage with is an exciting direction for future work!
> 2. **Meta-policy**:
>     - **Definition**: We define a meta-policy, $\beta: \mathcal{C} \mapsto p$ as a mapping from conversation prefixes to prompts.
>         - The PODP agent can first invoke $\beta$, and then the LLM with prompt $p\coloneq\beta(\mathcal{C})$ to produce a PODP action (i.e., a natural language response).
>         - In this case, the original problem of finding a good $\pi:\mathcal{C} \mapsto \mathcal{A}$ mapping is reduced to finding a good meta-policy $\beta:\mathcal{C} \mapsto p$, described by the equation at the end of Section 2.1.
>     - **Approach to learning meta-policy**:
>         - To clarify: we do not learn the meta-policy $\beta$ through online interactions---as you point out, this would require extensive interaction with humans.
>         - *If* we had a way to simulate the PODP environment, you are correct that we could use PPO to update the parameters of $\beta$. Instead, we use the *offline*, Q-learning-style approach outlined in Section 4.2 to learn $\beta$.
> 3. **Assumptions regarding offline data**:
>     - We agree that we need good data for the offline RL approach to work well. Rather than make strong data assumptions to argue for it, we instead devised a data-agnostic approach also (Section 4.1).
>     - We believe that both designing good prompts (Section 4.1) and learning meta-policies (Section 4.2) can be effective fixes in different data regimes.
>         - When historical data is not representative of future conversations, we may prefer Clarify-Flex over learning a meta-policy.
>         - Alternatively, if users engage exclusively in single-turn interactions, the baseline LLM responses may already be well-calibrated.
>     - We will include a discussion of the relative merits of our two proposed approaches in Section 4.
> 4. **Reward function for PODP and Q-learning settings**:
>    - **PODP setting**: The reward function of the PODP, $U$, takes the entire conversation as input to output a scalar, $U: \theta \times \mathcal{C} \mapsto \mathbb{R}$
>    - For example, many LLM-based chatbots allow users to rate their conversation with a thumbs up/down; these ratings can be directly interpreted as $U(\theta,\mathcal{C})$.
>        - Recent work (``Interpretable User Satisfaction Estimation for Conversational Systems with Large Language Models'', [Lin et al.](https://arxiv.org/abs/2403.12388)) suggests that $U$ can instead be inferred across a user population with a small sample of annotated conversations.
>     - **Q-learning setting**: In Section 4.2, we assume that we can access conversation logs that are annotated with their *trajectory-level* reward { $\mathcal{C}_i, U_i := U(\theta_i, \mathcal{C}_i) $ } $ _{i=1}^{N}$. We will revise Section 2.4 to describe the required training data more thoroughly.
>         - Picking a good prompt for each conversation turn is a *different* decision-making problem than the original PODP.
>         - Our approach is motivated by the Asymmetric Imitation Learning literature (``Asymmetric actor-critic for image-based robot learning'', [Pinto et al RSS'17](https://www.roboticsproceedings.org/rss14/p08.pdf)), where the teacher has full observability (e.g. the true $U(\theta,\cdot)$) and only the student suffers from partial observability (e.g. no access to $\theta$).

---

### Official Review · Reviewer_5dXh · 2024-03-21

**Q2-1 Originality-Novelty:** 4
**Q2-2 Correctness-Technical Quality:** 3
**Q2-5 Clarity Of Writing:** 4

**Q1 Summary And Contributions:**

The paper addresses the challenge of handling under-specified queries by LLM-based chatbots, especially in the context of recommendation systems. The authors identify that LLMs often struggle with under-specified queries, leading to responses that may not align with user intentions due to miscalibrated conversational priors. This misalignment is attributed to the training process of LLMs, where annotator biases and the nature of single-turn annotations may not accurately represent multi-turn conversation dynamics or user preferences in real-world recommender system interactions. The contributions of the paper are: The paper analyzes the query under-specification in real-world datasets and its impact on the performance of LLM-based chatbots. It proposes a formal framework for understanding under-specified queries as Partially Observed Decision Processes (PODPs) and develops interventions to recalibrate LLM responses to better handle such queries.
Through experiments using synthetic and real-world datasets, the authors demonstrate the effectiveness of their proposed interventions in improving the utility and relevance of chatbot responses in recommendation scenarios.

**Q2-3 Extent To Which Claims Are Supported By Evidence:**

3: Good: the main claims are supported by convincing evidence (in the form of adequate experimental evaluation, proofs, (pseudo-)code, references, assumptions).

**Q2-4 Reproducibility:**

3: Good: key resources (e.g. proofs, code, data) are available and key details (e.g. proofs, experimental setup) are sufficiently well-described for competent researchers to confidently reproduce the main results.

**Q3 Main Strengths:**

- The paper tackles the novel, often overlooked but significant issue of under-specified queries.
- The introduction of a formal framework (PODPs) for under-specified queries is a significant contribution, offering a structured way to analyze and address the challenges posed by query under-specification in conversational agents.
- The comprehensive experiments conducted with both synthetic and real-world datasets effectively demonstrate the prevalence of under-specification in user queries and the potential of the proposed interventions to improve LLM-based chatbot responses.
- The paper is very well-written.

**Q4 Main Weakness:**

The user modelling aspect needs to be studied more systematically. The paper assumes user willingness to engage in multi-turn interactions to clarify under-specified queries, which may not always be the case in real-world scenarios. User engagement and response rates to clarification prompts are not thoroughly examined.

**Q5 Detailed Comments To The Authors:**

The paper is well presented and easy to follow.

**Q9 Complying With Reviewing Instructions:**

Yes

---

> ### Author Rebuttal · Authors · 2024-04-09
>
> Thank you for your helpful feedback. Please see our response to your question below:
>
> 1. **User modeling assumptions**:
>    - We agree that in real-world settings, the users' willingness to answer clarifications may impact the realized gains associated with our proposed approaches.
>    - User studies have been conducted with search engines ("Analyzing and Learning from User Interactions for Search Clarification", [Zamani et al. SIGIR'20](https://arxiv.org/abs/2006.00166)) and pre-LLM conversation systems (``Towards Conversational Recommender Systems'', [Christokopoulou et al. KDD'16](https://dl.acm.org/doi/10.1145/2939672.2939746)) that demonstrated that users *do* engage with clarifying questions in those contexts.
>    - In our work, we diagnose query under-specification as a root cause of some sub-optimal LLM responses, and propose two intervention strategies. User studies of our approaches---and the viability of encouraging clarification in LLM chatbots in general---is an exciting direction for future work!

---

### Official Review · Reviewer_meqr · 2024-03-23

**Q2-1 Originality-Novelty:** 3
**Q2-2 Correctness-Technical Quality:** 2
**Q2-5 Clarity Of Writing:** 1

**Q10 Ethical Concerns:**

Not found.

**Q1 Summary And Contributions:**

This paper shows that the under-specification in human queries leads to undesired behavior from an LLM-based chatbot. Then, the authors proposed a taxonomy of response strategy types that allows us to analyze and improve miscalibrated policies from LLMs. Next, they provided a formal framing for under-specified queries as a Partially Observed Decision Process (PODP) where the aim of the chatbot policy is to maximize user utility with respect to partially observed goals. After that, they proposed two interventions to address the challenges described above and make a LLM-based chatbot produce appropriate conversational responses in scenarios despite underspecified user queries.

**Q2-3 Extent To Which Claims Are Supported By Evidence:**

1: Poor: the authors fail to convincingly backup their main claims (e.g., if the experimental evaluation is flawed, proofs are lacking or invalid, references are missing, assumptions are not realistic, not specified, or not motivated).

**Q2-4 Reproducibility:**

2: Fair: key resources (e.g. proofs, code, data) are unavailable but key details (e.g. proof sketches, experimental setup) are sufficiently well-described for an expert to confidently reproduce the main results.

**Q3 Main Strengths:**

The problem is interesting and the approach is good.

**Q4 Main Weakness:**

The paper is not well-written. There are several ambiguities in the paper. All results are given in appendix.

**Q5 Detailed Comments To The Authors:**

What are the under-specified requests? Are they out of domeain questions?

In equation (1), is the policy $\pi$ fixed for multi-turn system or is it changed?

What are differences among policies, meta-policies, custom-policies, and RLHF-policies?

How did you prove that policies are sub-optimal?

Is the proposed approach better than designing good prompts?

**Q9 Complying With Reviewing Instructions:**

Yes

---

> ### Author Rebuttal · Authors · 2024-04-09
>
> Thank you for your helpful feedback. Please see our responses below:
>
> ### **Re: weaknesses**:
> 1. **Ambiguity of presentation**:
>     - We apologize for the confusion. In Sec. 2, we frame user-chatbot interactions as a partially observable decision process (PODP), and in Sec. 4.2, we consider a *separate* decision-making task---that of learning a meta-controller using logged interaction data. We will revise notation to avoid inadvertent collisions when describing the two tasks.
> 2. **Empirical results in appendix**:
>     - All core empirical results are included in the main paper (i.e., Figs. 5-10). Many of our experiments rely on helper LLM tasks (ie, query and response classification), and we only include validation results of these helper LLM calls in the appendix (i.e., A.1; A.2).
>
> ### **Re: questions**:
> 1. **Under-specified queries**:
>     - Under-specified queries need not be out-of-domain questions. Rather, we consider a query *under-specified* if a human responder is likely to recognize that critical details (required to provide a high-quality answer) are missing. We use this insight to develop an LLM-based query underspecification classifier (see Appendix A.1 for prompt). We will include examples in the appendix of underspecified OpenAssistant queries we detected, here are two examples:
>         - "Suggest me places near 72nd St where I can park my car. Please also order them by price and add the price of each one to the right of their names."
>         - "I am not feeling that well, how can I know if I have sleep deprivation?"
> 2. **(Eq.1) is policy fixed in multi-turn?**:
>     - In a PODP the chatbot's policy, $\pi: \mathcal{C} \mapsto \mathcal{A}$ is a *fixed* mapping from conversation prefixes (which can span multiple turns) to natural language responses. While $\pi$ is stationary, it need not be Markovian because it takes the entire conversation trajectory as its input.
> 3. **Distinguishing between policy types and meta-policy**:
>     - **policy** (Eq. 1): We define the policy of a chatbot interacting with a user in the PODP setting as: $\pi: \mathcal{C} \mapsto \mathcal{A}$---i.e., a fixed mapping from conversation prefixes (which can span multiple turns) to natural language responses.
>     - **RLHF policy** (Eq. 3): When we implement a PODP policy by querying a blackbox LLM API with context $\coloneq \mathcal{C}$, we refer to the induced policy as the RLHF policy $\pi^{RLHF}$. We can expect good performance only if the RLHF-finetuned LLM guarantees that $\pi^{RLHF} \approx \pi^\ast$ (which is unverifiable).
>     - **custom policy** (Eq. 2): Here, we refer to policies derived by *prompting* LLMs---i.e., $\pi^p$, where the prompt, $p$, contains instructions that will be templated along with the conversation $\mathcal{C}$ to produce the LLM's context. $\pi^p \neq \pi^{RLHF}$ in general.
>     - **meta-policy** (Sec. 4.2): We define a meta-policy, $\beta: \mathcal{C} \mapsto p$ as a mapping from conversation prefixes to prompts.
>         - The PODP agent can first invoke $\beta$, and then the LLM with prompt $p\coloneq\beta(\mathcal{C})$ to produce a PODP action (i.e., a natural language response).
>         - The original problem of finding a good *policy* is thus reduced to finding a good *meta-policy*, as described by the equation at the end of Section 2.1.
> 4. **Suboptimality of baseline LLM**:
>     - **Qualitative evidence**: In Fig.5, we characterize default LLM behavior and note that even under severe under-specification, LLMs hedge (imposing an undue burden on the user, Fig.2) or respond directly (often making incorrect assumptions, Fig.3). Costly or incorrect responses are unlikely to be optimal in the PODP.
>     - **Empirical evidence**: In Figs. 9 and 10, we show that with a synthetic user model (that provides templatized answers to questions), it is possible to improve upon the performance of the baseline LLM---i.e., Clarify-Flex and a learned meta-policy perform better than $\pi^{RLHF}$ when evaluated in the PODP.
>     - We will improve Sec. 2.2 to discuss from first principles how RLHF produces sub-optimal policies for PODPs where high uncertainty regarding the belief state warrants/favors info-gathering actions.
> 5. **Learn meta-policy vs. design good prompts**:
>   - Our empirical results demonstrate that *both* strategies we evaluate---i.e., *designing good prompts* (Sec. 4.1), and *learning meta-policies* (Sec. 4.2) can be effective fixes in different data regimes:
>     - In our experiments with a synthetic user model, we observe (Fig. 9 and 10) that baseline LLM $<$ good prompt (Clarify-Flex) $<$ learned meta-policy.
>     - However, this ordering may not be universal: when historical data is not representative of future conversations, we may prefer Clarify-Flex over learning a meta-policy. We will add a discussion of the relative merits of each approach in Sec. 4.

---

### Meta-Review · Area_Chair_X4rv · 2024-04-19

The paper studies the calibration of LLMs to human communicative intent. They show that under-specified queries lead to undesired (low utility) interactions and that this is due to how the RLHF lab setting misrepresents realistic user interactions. The paper is clear and the various analyses rather insightful. The paper also contributes an approach for addressing this problem, framing interaction as a partially observed decision process. This approach is shown to be effective.

The paper approaches uncertainty as a feature (which, in my view, is the only reasonable way to approach uncertainty when natural language mediates interactions). We think the paper is clear, the topic is relevant, the work is complete and expect it to have some impact in this space.

This paper got a lot of praise, it inspired me to read it closely, and I agree with all the praise; I also disagree strongly with the one fairly negative review in the batch, and am choosing to down-weight its contribution to the recommendation (I suspect its criticism stems from misunderstandings, perhaps a mismatch in interests or background). I'm also bumping the two 'borderline accepts' by a point of two as they mostly asked clarification questions (listed as weaknesses) and, in my best assessment, the author response addressed those questions clearly and convincingly. I do suggest that the authors incorporate as much of the clarification points as possible.